First photographic records of the giant manta ray Manta birostris off eastern Australia

Couturier Lydie I.E. 1 2 lydie.couturier@outlook.com
Jaine Fabrice R.A. 2 3 4
Kashiwagi Tom 1 5
1 School of Biomedical Sciences, The University of Queensland , St Lucia , Australia
2 Climate Adaptation Flagship, CSIRO Marine and Atmospheric Research , Dutton Park , Australia
3 Biophysical Oceanography Group, School of Geography, Planning and Environmental Management, The University of Queensland , St Lucia , Australia
4 Marine Megafauna Foundation, Manta Ray & Whale Shark Research Centre , Tofo Beach, Inhambane , Mozambique
5 Molecular Fisheries Laboratory, The University of Queensland , St Lucia , Australia
Bruno John
Electronic publication date: 2015 Jan 22
Publication date: 2015
Volume: 3
Electronic Location ID: e742
Received 2014 Oct 31; Accepted 2015 Jan 5
Copyright: © 2015 Couturier et al.
Copyright year: 2015
Copyright holder: Couturier et al.
License: This is an open access article distributed under the terms of the Creative Commons Attribution License, which permits unrestricted use, distribution, reproduction and adaptation in any medium and for any purpose provided that it is properly attributed. For attribution, the original author(s), title, publication source (PeerJ) and either DOI or URL of the article must be cited.
License URL: https://creativecommons.org/licenses/by/4.0/

Keywords: Mobulidae, Manta ray, Australia, Citizen science

Funding: Australian Research Council Linkage Projects LP110100712 Manuscript preparation was partially supported by Project Manta, a research program based at the University of Queensland, funded and supported by Australian Research Council Linkage Projects Grant LP110100712, Earthwatch Institute Australia, Lady Elliot Island Eco Resort and Manta Lodge Scuba Centre. The funders had no role in study design, data collection and analysis, decision to publish, or preparation of the manuscript.

==============================
We present the first photographic evidence of the presence of the giant manta ray Manta birostris in east Australian waters. Two individuals were photographed off Montague Island in New South Wales and off the north east coast of Tasmania, during summer 2012 and 2014, respectively. These sightings confirm previous unverified reports on the species occurrence and extend the known distribution range of M. birostris to 40°S. We discuss these findings in the context of the species’ migratory behaviour, the regional oceanography along the south east Australian coastline and local productivity events.

Introduction

Manta rays (Manta spp.) are amongst the largest filter-feeding elasmobranch fishes and have a circumglobal distribution through tropical and temperate coastal waters, offshore islands and seamounts (Marshall, Compagno & Bennett, 2009). Manta rays belong to the family Mobulidae, comprising the two genera Manta Bancroft, 1829 and Mobula Rafinesque, 1810. All mobulid species are epipelagic zooplanktivores that are presumed to be long lived (e.g., >30 years for Manta spp.) and have low fecundities (i.e., late maturity, long gestation period and only a single large pup) (Couturier et al., 2012). Previously considered to be monospecific (Manta birostris), the genus Manta was redescribed in 2009 to comprise two distinct species: the reef manta ray Manta alfredi (Krefft, 1868) and the giant manta ray Manta birostris (Walbaum, 1792), and a third putative species M. cf. birostris (Marshall, Compagno & Bennett, 2009). Both recognised species have circumglobal distributions, sympatric in some areas and allopatric in others (Kashiwagi et al., 2011). Manta birostris is considered a more oceanic and migratory species, and is found predominantly in cooler, temperate to subtropical waters (Marshall et al., 2011). Manta alfredi displays a high degree of site fidelity in tropical and subtropical waters, but may also undertake local to regional-scale (>700 km) movements and seasonal migrations (Dewar et al., 2008; Couturier et al., 2011; Deakos, Baker & Bejder, 2011; Marshall, Dudgeon & Bennett, 2011; Couturier et al., 2014; Jaine et al., 2014).

Both manta ray species and four of the nine described Mobula species are reported to occur in tropical to temperate waters of Australia (Last & Stevens, 2009; Marshall, Compagno & Bennett, 2009). While the occurrence of M. alfredi has been widely confirmed off the coast of eastern Australia (Couturier et al., 2011; Couturier et al., 2014), the occurrence of M. birostris in this region has been lacking photographic validation despite records in literature (Hutchins & Swainston, 1986; Allen et al., 2006; Last & Stevens, 2009). The recent separation in the genus Manta spp. means that records of M. birostris prior to 2009 lacking photographic evidence cannot be validated, as species may have been confused with M. alfredi. This paper presents the first photographic evidence confirming the occurrence of M. birostris in east Australian waters, with one specimen photographed off Montague Island, New South Wales, in January 2012 and one specimen photographed off the northeast coast of Tasmania in January 2014.

Materials and Methods

As part of a larger study, photographs of manta rays were sought from dive clubs, dive instructors, researchers and recreational divers along eastern Australia for photographic identification purposes (see Couturier et al., 2011). Four photographs and two video recordings of a free swimming M. birostris were taken by Peter McGee, a recreational diver, off Montague Island (36°15′7.15″S; 150°13′35.19″E; Fig. 1) offshore from Narooma in southern New South Wales (Specimen #1, Fig. 2). The individual was sighted near an Australian fur seal Arctocephalus pusillus (Schreber, 1775) colony on the 5th January 2012, swimming at a depth of about 13 m, in 21 °C waters (P McGee, pers. comm., 2013).

Figure 1 Map of south east Australia showing sighting locations of specimen #1 (Montague Island) and specimen #2 (north east Tasmania).

One photograph of a free swimming M. birostris was taken from the surface by Leo Miller, a recreational fisherman, off the north east coast of Tasmania (40°S; 148°E, no precise location given; Fig. 1) on the 26th January 2014. The photograph was submitted to the University of Tasmania Institute of Marine and Antarctic Studies’ Redmap website http://www.redmap.org.au/ (Specimen #2, Fig. 3).

Figure 2 Photographs of a Manta birostris (specimen #1) taken off Montague Island on the 5th January 2012 by Peter McGee.

White arrows indicate key characters for M. birostris as described in Marshall, Compagno & Bennett (2009): (A) and (B) show distinctive dorsal coloration with white shoulder patches with their anterior margins extending medially from spiracles in an approximately straight line parallel to the edge of the mouth; (C) and (D) show large semi-circular spots posterior to the fifth gill slits and grey V-shaped margin along posterior edge of the pectoral fin; and (D) shows dark coloration around mouth and the caudal spine, embedded in a calcified mass and covered with a skin layer, immediately posterior to the dorsal fin (white box).

Figure 3 Photographs of a Manta birostris (specimen #2) taken off the north east coast of Tasmania on the 26th January 2014 by Leo Miller.

White arrows indicate distinctive dorsal white shoulder patches with their anterior margins extending medially from spiracles in an approximately straight line parallel to the edge of the mouth, as key character of the dorsal colouration of M. birostris as described in Marshall, Compagno & Bennett (2009).

Characters used to confirm identification of Manta spp. were terminal mouth, broad head and body coloration. Species identification was based on key morphological features provided by Marshall, Compagno & Bennett (2009), including (1) distinct shoulder patches with triangular shape, (2) presence of a caudal spine, (3) distinctive dark spots on the ventral side over abdominal region, with no spots present medially between the gill slits, (4) prominent semi-circular marking extending posteriorly from both 5th gills and (5) dark-coloured margin on posterior edges of pectoral fins.

Results and Discussion

Key morphological features, including terminal mouth, broad head, distinctive ventral and dorsal coloration, and presence of caudal spine, could be distinguished from photographs of Specimen #1 (Fig. 2). Together these features allow the specimen to be identified as M. birostris and positively differentiated from M. alfredi, also known to occur in east Australian waters (Couturier et al., 2011). The distinctive dorsal coloration of Specimen #2 was the only observable morphological feature identifying this individual as M. birostris (Fig. 3).

The occurrence of M. birostris off Montague Island at ∼36°S in east Australia is consistent with records in south western Pacific Ocean where the species occurs up to 36°S (Duffy & Abbott, 2003; Kashiwagi et al., 2011) and in the south western Atlantic where it occurs up to 34°S (Marshall et al., 2011). Manta ray sightings off Montague Island have been reported in a scuba divers guide (Byron, 1986) and in anecdotal reports (N Coleman & J Van Der Westhuizen, pers. comm., 2012). Manta rays are also commonly advertised as possible diving encounters during austral summer by most dive operators using this dive site (e.g., Narooma Charters, Islands Charters). These unverified sightings were likely to be of M. birostris considering that M. alfredi distribution range does not appear to extend beyond 30°S worldwide (Marshall, Compagno & Bennett, 2009; Couturier et al., 2014). In addition, M. alfredi was not sighted southward of the South Solitary Island (30°12′24.33″S, 153°16′2.52″E) in east Australia despite continuous monitoring effort along the coast over the last 5 years (Couturier et al., 2011; Couturier et al., 2014).

The scarce information available on M. birostris migratory ecology suggests that its movements are timed with seasonal oceanographic events known to enhance productivity. Seasonal occurrence of the species off south-eastern Brazil was associated with a low salinity coastal front (Luiz et al., 2009), while movements of tagged manta rays in the Gulf of Mexico were linked to seasonal upwelling events and thermal fronts (Graham et al., 2012). Manta birostris and several Mobula spp. also occur off North East New Zealand during summer months, which coincide with the path and flow of the East Auckland Current (Duffy & Abbott, 2003).

The occurrence of M. birostris off Montague Island may be linked to regional circulation patterns and productive oceanographic events during summer. The East Australian Current (EAC) flows pole-ward along the east Australian coast and its main EAC jet bifurcates abruptly to the east at ∼32°S. About a third of the main EAC jet continues south into the Tasman Sea, towards Montague Island, as a series of dynamic eddies (Ridgway & Godfrey, 1997; Roughan et al., 2011). Enhanced nutrient concentrations and upwelling processes have been documented during austral spring and summer south of the separation point where Montague Island is located (e.g., Oke & Middleton, 2001; Roughan & Middleton, 2004; Ridgway, 2007). These conditions generate ephemeral but highly productive phytoplankton blooms along the coast (Hallegraeff & Jeffrey, 1993; Bax et al., 2001), that likely boost the abundance of zooplankton prey. Humpback whales Megaptera novaeangliae regularly feed on small pelagic fish and coastal krill Nyctiphanes australia along the southeast Australian coast during their southward migration (Stamation et al., 2007). It is probable that M. birostris also occur in this area during warmer months to exploit local productivity events.

The occurrence of M. birostris off north east Tasmania at ∼40°S is the newly-extended southern-most record for the species. This sighting may be linked to exceptional oceanographic conditions occurring in the area at the time of the sighting or a response to warming waters by climate-driven changes. South-east Australia is a global warming hotspot where the sea surface temperatures have been increasing up to 3 times the global average rate over the past 50 years, and are projected to rise further into the future (Ridgeway & Hill, 2012; Wu et al., 2012; Hobday & Pecl, 2014; Oliver et al., 2014). Southward range extensions have already been reported in this region for plankton communities, macroalgae, macro-invertebrates and fish (Johnson et al., 2011; Last et al., 2011; Ridgeway & Hill, 2012). Sea surface temperatures around the sighting area usually vary between 12 °C in winter and 17 °C in summer (Condie & Dunn, 2006). In warm years, temperatures were reported to increase up to 2 °C above average temperatures recorded 60 years ago due to circulation changes of the EAC (Ridgway, 2007; Ridgeway & Hill, 2012). Although M. birostris may tolerate low temperature for short periods of time (e.g., during deep dives), its distribution in tropical and subtropical waters suggest a preference for temperatures above 17 °C (Marshall et al., 2011). It is possible that at the time of the sighting the EAC flow had extended southward along the Tasmanian coast with increased strength (Ridgeway & Hill, 2012; Oliver & Holbrook, 2014), engendering favourable environmental conditions for M. birostris. In addition to providing a suitable thermal habitat, the intrusion of warmer waters along the east Tasmanian coast may trigger productivity events (Matear et al., 2013), providing new food resources for the species.

Based on these observations, we confirm the presence of M. birostris for the first time in east Australian waters, increasing the known range of the species. The scarcity of recorded observations of M. birostris compared to M. alfredi, despite vibrant diving and boating activities along the ∼4,000 km east Australian coastline, suggests that the species is rare in the area. It is also possible that the species occupies and traverses areas that are not exploited by fisheries and/or tourism and thus remain undetected.

The authors wish to thank the two photographers Peter McGee and Leo Miller for contributing photographs, and J Bruno, R Graham and S Prado for their valuable comments on the manuscript.

Additional Information and Declarations

Competing Interests

Author Contributions

Lydie I.E. Couturier and Fabrice R.A. Jaine are employees of CSIRO Marine and Atmospheric Research, and Fabrice R.A. Jaine is an employee of the Marine Megafauna Foundation.

Lydie I.E. Couturier conceived and designed the experiments, performed the experiments, analyzed the data, contributed reagents/materials/analysis tools, wrote the paper, prepared figures and/or tables, reviewed drafts of the paper.

Fabrice R.A. Jaine performed the experiments, contributed reagents/materials/analysis tools, wrote the paper, prepared figures and/or tables, reviewed drafts of the paper.

Tom Kashiwagi contributed reagents/materials/analysis tools, wrote the paper, reviewed drafts of the paper.

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
