# Peer review of "First photographic records of the giant manta ray Manta birostris off eastern Australia"

_PeerJ, doi:10.7717/peerj.742_

## Round 0.1 · original submission · Minor Revisions

Please add the natural history information the reviewers suggested, including migratory ecology, diet, etc.

·

Basic reporting

This short manuscript provides compelling evidence and rationale regarding the presence of M. birostris in eastern Australian waters.

As a manuscript documenting the occurrence of a species based on two incidental sightings, the background information, the identification of M. birostris based on the photographs, and the context of the findings are appropriate.

The one main issue is the lack of discussion on the migratory ecology of M. birostris (as it is advertised in the abstract) and should be added accordingly. Suggested references could include:

Ridgway, K.R. (2007). Long-term trend and decadal variability of the
southward penetration of the East Australian Current. Geophys. Res.
Lett., 34.

Hill, K.L., Rintoul, S.R., Coleman, R. & Ridgway, K.R. (2008). Wind
forced low frequency variability of the East Australia Current.
Geophys. Res. Lett., 35.

Hobday, A.J. & Pecl, G.T. (2014). Identification of global marine
hotspots: sentinels for change and vanguards for adaptation. Rev. Fish
Biol. Fish., 24, 415–425.

Holbrook, N.J. & Bindoff, N.L. (1997). Interannual and Decadal
Temperature Variability in the Southwest Pacific Ocean between 1955
and 1988. J. Clim., 10, 1035–1049.

Oliver, E.C.J. & Holbrook, N.J. (2014). Extending our understanding of
South Pacific gyre “spin-up”: Modelling the East Australian Current in
a future climate. J. Geophys. Res. – Ocean., early onli

Experimental design

No hypothesis being tested, so not applicable.

Validity of the findings

No comments.

Additional comments

I must say I am surprised that there weren't any prior photographic records of this species!

The only topic that needs to be addressed is the discussion on the migratory ecology of M. birostris. It was mentioned in the abstract that migration was part of the manuscript, yet in the rest of the paper the only sentence that could be related to migration is that dive operators advertise potential manta ray encounters during the austral summer. If it is known that oceanic manta rays only reach these southern latitudes in summer, it needs to be stated clearly. Even if the migration patterns of M. birostris are completely unknown, it should also be mentioned.

Specific comments:

Abstract
Given that people reading the abstract haven't read the paper yet and don’t necessarily know the distribution of M. birostris, it needs to be made clear that the range extension occurs southwards to 40°S.

Line 36. Remove “and potentially Mobula spp”. The sentence is about the recent split in the genus Manta, however confusing mantas with mobulas is a problem regardless of split.

Lines 45. The word “off” is repeated twice in a row. I suggest removing “off Narooma” or replacing with “offshore from Narooma”.

Line 51. Remove “the” before “26th January 2014”.

Line 53. Remove colon after were to improve readability.

Line 59. I don’t see how the systematics are relevant for this paper. given that this paper is not about taxonomy, I suggest removing this from the manuscript for the sake of brevity.

Line 71. Remove “and” after “M. alfredi”.

Line 77. Are these manta ray sightings off Montague Island more likely to be M. birostris or M. alfredi? Specifying if this is known or not would help the reader.

Line 95. What is it meant by a global warming hotspot? Being specific about what this means would help drive your point across. How many degrees warming does an area have to undergo to be considered a hotspot? How many °C higher than average does this area experience?

Line 98. The confirmation of the sightings as M. birostris were done based on observations by the authors of the paper as well as on those of the two photographers. I suggest replacing “Based on our observations” with “Based on the previous observations”.

Figure 1
Given that latitudes are mentioned throughout the manuscript because of the southward range expansion, it would be useful to have these on the map as well.

Figure 2
Line 180. Replace “shows” with “show”. Also (d) is mentioned twice and reads awkwardly. Make this legend clear.

·

Basic reporting

This paper adheres to the PeerJ policies and structure and although is not meant to test a hypothesis provides a valid insight into the range extension of Manta birostris.

Experimental design

No Comments.

Validity of the findings

The authors' findings shed additional information on the world's largest ray, and should be published. The manuscript should however include additional information on known prey species occurring at the sites during the observation periods as well as general temperatures preferred by M. birostris. Information on the temp and depth during the second sighting and the most annual temperature ranges recorded at both observation sites, notably at 40 degrees S should also be included. Published insights or linkages to "exceptional oceanographic conditions occurring in the area at the time of the sighting" are recommended. These will help to better place this paper's findings in the context of known Mb ecology.

---

## Round 0.2 · accepted · Accept

I appreciate your careful responses to the suggested edits by the two reviewers.